# Bispecific Antibodies: A Novel Approach for the Treatment of Solid Tumors

**DOI:** 10.3390/pharmaceutics14112442

**Published:** 2022-11-11

**Authors:** Luigi Liguori, Giovanna Polcaro, Annunziata Nigro, Valeria Conti, Carmine Sellitto, Francesco Perri, Alessandro Ottaiano, Marco Cascella, Pio Zeppa, Alessandro Caputo, Stefano Pepe, Francesco Sabbatino

**Affiliations:** 1Department of Clinical Medicine and Surgery, University of Naples “Federico II”, 80131 Naples, Italy; 2Oncology Unit, Department of Medicine, Surgery and Dentistry, University of Salerno, 84081 Baronissi, Italy; 3Clinical Pharmacology Unit, Department of Medicine, Surgery and Dentistry, University of Salerno, 84081 Baronissi, Italy; 4Medical and Experimental Head and Neck Oncology Unit, INT IRCSS, Foundation “G. Pascale”, 80131 Naples, Italy; 5SSD Innovative Therapies for Abdominal Metastases, Abdominal Oncology, INT IRCCS Foundation “G. Pascale”, 80131 Naples, Italy; 6Unit of Anesthesiology and Pain Therapy, INT IRCCS Foundation “G. Pascale”, 80131 Naples, Italy; 7Pathology Unit, Department of Medicine, Surgery and Dentistry, University of Salerno, 84081 Baronissi, Italy

**Keywords:** antibodies, bispecific, bsAb, clinical trials, immunotherapy, mAb, solid malignancies

## Abstract

Advancement in sequencing technologies allows for the identification of molecular pathways involved in tumor progression and treatment resistance. Implementation of novel agents targeting these pathways, defined as targeted therapy, significantly improves the prognosis of cancer patients. Targeted therapy also includes the use of monoclonal antibodies (mAbs). These drugs recognize specific oncogenic proteins expressed in cancer cells. However, as with many other types of targeting agents, mAb-based therapy usually fails in the long-term control of cancer progression due to the development of resistance. In many cases, resistance is caused by the activation of alternative pathways involved in cancer progression and the development of immune evasion mechanisms. To overcome this off-target resistance, bispecific antibodies (bsAbs) were developed to simultaneously target differential oncogenic pathway components, tumor-associated antigens (TAA) and immune regulatory molecules. As a result, in the last few years, several bsAbs have been tested or are being tested in cancer patients. A few of them are currently approved for the treatment of some hematologic malignancies but no bsAbs are approved in solid tumors. In this review, we will provide an overview of the state-of-the-art of bsAbs for the treatment of solid malignancies outlining their classification, design, main technologies utilized for production, mechanisms of action, updated clinical evidence and potential limitations.

## 1. Introduction: From Monoclonal Antibodies to Bispecific Antibodies

In the last two decades, monoclonal antibody (mAb)-based therapy has remarkably changed the therapeutic landscape of several human diseases including solid malignancies [1]. Structurally, mAbs are two heterodimers composed of two identical heavy and light chains. The light chain has one variable (VL) and one constant (CL) domain, whereas the heavy chain has one variable (VH) and three constant (CH; CH1, CH2, CH3) domains [2]. In addition, each mAb contains a Fragment Ab-binding (Fab) region and one Fragment Crystallizable region (Fc) [3]. The Fab region is responsible for binding to unique epitopes [4], with agonistic, antagonistic or neutral properties, while the Fc region interacts with the Fc receptor expressed on immune cells and promotes various effector functions [5,6]. Through both the Fab and Fc regions, mAbs can exert anti-cancer effects in different ways: (i) antibody-dependent cellular phagocytosis (ADCP); (ii) complement-dependent cytotoxicity (CDC); (iii) antibody-dependent cell-mediated cytotoxicity (ADCC); (iv) induction of apoptosis; and (v) engagement of cell surface receptors with consequent inhibition or activation of signaling pathways [5,6] (Figure 1).

Over the years, several mAbs have been developed for the treatment of solid malignancies, improving the clinical outcomes of cancer patients [7]. For instance, mAbs targeting Epidermal Growth Factor Receptor (EGFR) or Human Epidermal Growth Factor Receptor 2 (HER2) have dramatically improved the survival outcomes of patients affected by lung cancer, breast cancer and other solid malignancies [8,9]. However, mAb-based therapy usually fails in the long-term control of cancer progression due to the development of resistance [10]. In many cases, resistance is caused by the activation of alternative pathways involved in cancer progression [10]. In order to counteract tumor resistance, the implementation of novel agents targeting the involved alternative pathway components becomes a challenge [10]. Bispecific antibodies (bsAbs) display the potential to achieve this goal [11]. These molecules simultaneously target independent epitopes on two different antigens, determining the blockade of mutually related signaling pathways [12]. This “double blockade” improves anti-cancer effects as compared to the single blockade [13]. In addition, the resistance to mAb-based therapy can also be mediated by the development of immune escape mechanisms [10], which allow cancer cells to evade the host cancer immune response [10]. Even in this case, bsAbs can improve cancer cell elimination by promoting the interaction of cancer cells with cognate T cells in the tumor microenvironment (TME) [11,13].

Production of bsAbs requires two different heavy chains and two different light chains [2,13]. The differential pairing of heavy and light chains from two antibodies can theoretically result in 16 potential differential combinations (10 different molecules), one bispecific and the others both non-functional and monospecific [14]. So far, different strategies have been developed in order to overcome the limitations linked to functional pairing in bsAb generation [14]. The first study testing the production of antibodies with mixed specificities was initiated in 1960 by Alfred Nisonoff [15]. Later, in 1985, Staerz et al. introduced the hybridoma technology to produce bsAbs [16]. This event, from 1985 to 1995, started a novel period called the “bispecific explosion” with a growing development of bsAbs [14,17]. Successively, another three methods were developed: (i) quadroma technology; (ii) chemical conjugation-based methods; and (iii) recombinant DNA-based methods [14,18]. All these methods have revolutionized the scenario of bsAb development, enabling researchers to modulate some important features of bsAbs such as size, valency, flexibility, half-life and biodistribution [12,14]. As a result, in the past two decades, over 100 different formats have been developed [12]. 

This novel therapeutic approach based on bsAbs represents a growing area of research [19]. Several clinical trials were implemented to test the efficacy and the safety of bsAbs, especially for the treatment of hematological malignancies [20,21]. Blinatumomab was the first bsAb approved by the Food and Drug Administration (FDA) for the treatment of Philadelphia chromosome-negative acute lymphoblastic leukemia (ALL). Later, other bsAbs were also approved for the treatment of other hematological malignancies [22,23]. On the other hand, contrasting results were shown in the implementation of bsAb-based therapy for the treatment of solid tumors [13]. 

The aim of this review was to provide a global overview of types of bsAb classification, design, production methods, mechanisms of action, and clinical staging of efficacy evaluation. The focus of the work was to provide an up-to-date account of the most noteworthy clinical trials in which bsAbs were tested or are being tested for the treatment of solid malignancies. 

## 2. Classification and Production of Bispecific Antibodies

The “zoo” of bsAbs is populated by many different species, comprising around 100 different “formats”, classified according to their structure and mechanisms of action (MoA) [24]. In addition, bsAbs can be categorized based on the number of recognized antigens (specificity), the total number of antigen-binding sites per molecule (valency), and interaction strength between an epitope on the antigen and a paratope on the antibody (affinity) [12]. Structurally, bsAbs can be divided into two groups of immunoglobulin G (IgG)-like and non-IgG-like molecules based on the presence/absence of the Fc region, which determines different features in terms of functions as well as pharmacodynamic and pharmacokinetic profiles [24] (Figure 2).

### 2.1. Fc-Based bsAbs

Fc-based bsAbs (IgG-like molecules) are composed of homo- or heterodimeric Fc domains linked to Fc-free bsAbs by a peptide linker [25,26,27] (Figure 2). These bsAbs are characterized by a longer half-life as compared to Fc-free bsAbs [28] and a stronger ability to trigger effector functions such as ADCC, ADCP and complement fixation [5,13,29]. Fc-based bsAbs were generated for the first time by a quadroma technology that utilizes the somatic fusion of two different hybridoma cell lines [14,30]. Each hybridoma cell line expresses a murine mAb with its own specificity [14,30]. As a result, the quadroma cell line produces IgG-like bsAbs with two different arms and two specificities [14,30]. Nevertheless, during bsAb production, the quadroma cell line also produces non-functional antibodies because of the random pairing of light and heavy chains from two distinct antibodies [14,31]. Non-functional antibodies severely reduce the production yields [14,31]. In order to counteract this inefficient random chain pairing, chimeric quadroma cell lines were developed by combining both rat and murine hybridoma cell lines [14,32]. This methodology displays several advantages including (i) the enrichment of functional bsAbs by preferential specie-restricted heavy/light chain pairing (observed in four of four rat/mouse quadromas) as compared to the random pairing of conventional mouse/mouse or rat/rat quadromas, and (ii) the possible one-step purification of the quadroma supernatant [32]. Later than quadroma technology, chemical conjugation-based methods were implemented to generate Fc-based bsAbs (Figure 2). In order to better classify this type of Fc-based bsAbs, some authors provided an additional sub-classification into symmetric (IgG-like) and asymmetric (IgG-modified structures) bsAbs based on additional binding sites [33]. Specifically, the symmetric bsAbs derived from the fusion of a pair of identical polypeptide chains or paired light and heavy chains [14,33]. Although these types of bsAbs are relatively close to natural mAbs, these bsAbs differ from the latter in several features such as molecular size, structure, stability, and solubility. These features may influence pharmacodynamic and pharmacokinetic properties [14,33]. Technology platforms such as Dual Variable Domain Immunoglobulin (DVD-Ig™) and two-in-one are all symmetrical models [14,33]. The DVD-Ig™ approach was recently developed to generate bsAb format made of two different light and heavy chains [14]. Both light and heavy chains display an additional variable domain [14,34]. A specific linker sequence connects the additional variable domain to the VL and VH domains of the mAb [14,34]. The obtained bsAb displays four antigen recognition sites with each single Fab binding to two different targets [14,34]. By avoiding the mispairing of different light or heavy chains as well as by using a pair of different mAbs, the DVD-Ig™ approach allows for (i) modulating the specificity and valance of mAb, (ii) improving the production yield, and (iii) increasing the homogeneity and stability of the bsAb [14]. Two-in-one technology was developed in order to obtain bsAbs capable of recognizing two different targets called Dual Action Fab (DAF). DAF bsAbs are relatively similar to a normal IgG mAb showing a high level of stability and ease of industrial production [14,35]. Specifically, the recognition of a second antigen in addition to the first requires a mutation into the antigen-binding site [14,35]. Further engineering within the variable domain is required to achieve maximum dual affinity [14,35]. On the other hand, asymmetric format, in greater numbers than symmetric patterns, destroys the symmetrical structure promoting heavy chain heterodimerization of the bsAbs that can combine two different but complementary heavy chains through the engineering of the Fc region. Commonly used technology includes Knobs-into-Holes (KiH) [36]. Specifically, the “Knobs” are created by substituting small amino acids with big ones at the interface across the CH3 domain in one heavy chain, while the “holes” are created by substituting large amino acids with small ones in the other heavy chain [36]. Although this method promotes the correct heavy chain heterodimerization, it cannot prevent the random pairing of the two different heavy chains with the light chains [14,36]. As a result, alternative approaches were developed. One approach involves the purification of bsAbs from the media of two co-culture stable cell lines [14,37]. Half-antibodies are secreted and recombined to form homodimers and heterodimers [14,37]. Further addition of reduced glutathione helps disulfide formation and correct pairing [14,38]. Another approach utilizes the principle of transient transfection [14,39]. Specifically, plasmids for each heavy and light chain fragments are transfected into two mammalian cell lines and separately cultured. Two independent half-antibody molecules are secreted into the media. Antibody folding and assembly are accomplished by in vitro incubation of reduced glutathione in order to catalyze disulfide bond formation [14,39]. A third approach utilizes light and heavy chains of one half-antibody designed with complementary mutations in CH1-CL and VH-VL interfaces [14,39]. It results in a better pairing between two desired chains after transfection in mammalian cells [14,39]. Lastly, an alternative approach utilizes the transfection of *E. coli* with half-antibody plasmids [14,40]. During cell lysis, the heavy and light arm fragments dimerize and inter-chain disulfide bonds are formed. Following cell lysis, the assembled bsAbs are purified from the supernatant [14,40]. However, the lack of mammalian glycosylation modifications can affect the functions of the produced antibody [14,41].

Used in combination with the KiH technology, CrossMAb technology ensures an accurate light chain pairing during the bsAb assembly [14]. Specifically, produced bsAbs have one modified arm (both light and heavy chains) and an unmodified arm [14,42]. Modifications are limited either to CL-CH1 and VL-VH domains or to the entire Fab region [14,42]. As a result, the unmodified heavy chain can no longer associate with the modified light chain, enforcing the required chain association [14,43]. Among the potential modifications, the best composition and purity are achieved by CL-CH1 CrossMAb [14,43].

### 2.2. Fc-Free bsAbs

Fc-free bsAb (non-IgG-like molecules) represent a group of constructs lacking the Fc region generated by recombinant DNA-based methods (Figure 2). This group includes Fabs, single-chain variable fragments (scFvs) and single-domain antibodies (sdAbs) [44]. Specifically, the scFv consists of two variable domains connected by a peptide flexible linker (disulfide bonds) and/or noncovalent inter-domain interactions that originate from different mAbs [44]. In contrast, sdAb only contains a single variable domain (12–15 kDa), such as nanobodies derived from camelid heavy-chain antibodies (HCAbs) that include only heavy chains (VHHs). They display the smallest molecular weight (12–15 kDa) [14,25,26,27,44,45,46,47]. Fc-free bsAbs present several advantages as compared to Fc-based bsAbs. They include the efficient expression in a wide range of hosts (e.g., bacteria and mammalian cells) and the preservation of the binding activity of the parental antibody [44]. In addition, in terms of pharmacodynamic and pharmacokinetic profiles, Fc-free bsAbs display higher tumor penetration, better epitope accessibility, less immunogenicity, and lower incidence of immune-related adverse events (irAEs) [6,24,48]. On the other hand, Fc-free bsAbs also display some drawbacks as compared to Fc-based bsAbs. Involvement of the Fc region in antibody effector functions including ADCC, ADCP, and CDC represents an important characteristic of the Fc region in Fc-based bsAbs. Thus, Fc-free bsAbs lacking Fc-mediated effector functions might negatively affect their therapeutic activity in eradicating tumor cells through the activation of cognate T cells [13,29]. Second, stability and aggregation can represent another issue of Fc-free bsAbs. As a result, re-engineering of the end-products is often required [12]. Lastly, the small size of Fc-free bsAbs can cause poor retention time in the target tissue and rapid blood clearance. Thus, different strategies were employed in order to improve their serum half-life, including long-time continuous intravenous (cIV) infusion and/or specific structural modifications such as polyethylene glycosylation or fusion to human serum albumin as well as to an Fc part of an IgG molecule (IgG-like molecules) [28]. According to these limitations, based on the implementation of genetic engineering and the emergence of different methods, different Fc-free bsAbs were designed and developed including tandem scFvs, diabodies, tandem diabodies (TandAbs) and dual affinity retargeting (DART) proteins.

Tandem scFvs are bsAbs created by the fusion of two scFvs derived from different mAbs, linked by a long peptide that enhances the flexibility of the two different Antigen-binding sites (50–60 kDa) [14]. The format of tandem scFvs can be distinguished from others either by the direction of the VL and VH domains in the scFvs or by the connecting linker sequence [14]. In the past few years, tandem scFvs have been especially utilized for the development of bispecific T-cell engagers (BiTEs) usually produced from mammalian cells, presumably because of their rather complex tertiary structure and multiple disulfide bonds [14,21,49,50,51]. The latter consists of two scFvs, one binds to the CD3 on T cells, the other to the tumor antigen. As a result, cancer cells are linked to cytotoxic T cells [50] triggering a cascade of events including T cell activation, secretion of perforin and granzyme B, the release of cytokines engaging other types of immune cells and finally, tumor cell apoptosis induction [50]. Interestingly, BiTEs lead to the recognition and lysis of cancer cells by cytotoxic T cells even in the absence of major histocompatibility complex class I (MHC-I) expression [21,50]. Their compact size is essential to keep an optimal distance between the target and effector cells in order to favor cell killing [14,21,50,51,52,53]. On the other hand, their small size causes their rapid in vivo clearance. As a result, alternative structural strategies along with cIV infusion are required to prolong the half-life and provide optimal serum concentrations of BiTEs [14,21,50,51]. Stability represents another limitation for a subset of BiTE administration. While some BiTEs can be refrigerated for months in PBS without losing their binding activity [14,21,50,51,52,53], most BiTEs are stable for a maximum of 48 h at room temperature [14,21,50,51,52,53]. Despite these technical issues, some BiTEs such as blinatumomab, demonstrated superb therapeutic efficacy and several BiTEs are currently being investigated for the treatment of various hematological and solid malignancies [14,21,50,51,52,53]. 

Diabodies are compact bsAbs obtained by decreasing the size of the peptide linker of about five amino acids between the variable domains. This in turn favors the correct pairing of the domains from two different polypeptides, one of which contains the VH of the mAbA linked to the VL of the mAbB, and the other one contains the VH of the mAbB connected with the VL of the mAbA [44]. Similar to BiTEs, diabodies have two different antigen-binding sites [14,18]. In addition, mutations are introduced into the VL–VH interface to favor heterodimerization over homodimerization and an interdomain disulfide bond is engineered into the structure to improve the stability of the produced diabody [14,54]. Because of the instability of diabodies caused by incorrect dimers generated in the cell, different formats of diabody are developed to improve their stability such as the single chain diabody (scDb). Specifically, scDbs are the same size as tandem scFvs [14,54,55,56]. However, scDbs differ in the length of their linkers as well as in the arrangement of the domains [14,54,55,56]. In addition, scDbs display a more compact and less flexible structure as compared to tandem scFvs [14,54,55,56]. To restore the Fc-mediated biological activity and/or to extend the half-life, scDbs are fused with additional Fc or CH3 domains [14,54,55,56]. 

Based on scDb format, the tandem diabodies (TandAbs), a dimeric molecule with four binding sites, are designed [12]. The TandAb molecules (~114 kDa), which contain four antigen-binding sites, exhibit a prolonged half-life and higher binding affinity to the targets as compared to tandem scFvs and diabodies [12]. 

The other diabody-based bsAbs are the DART proteins. They contain two polypeptide chains linked by a disulfide bond in a VLmAbA-VHmAbB/VLmAbB-VHmAbA configuration in order to improve stability and easy manufacturability without a significant increase in aggregation [12,44]. 

## 3. Mechanisms of Action of bsAbs

Because bsAbs have two simultaneous binding specificities, their targeted pathways are quite flexible [12]. bsAbs are designed to exert anti-cancer effects through different mechanisms: (i) bridging cancer and immune cells for redirected cytotoxicity; (ii) promoting immune cell functions such as T-cell expansion and release of granzymes and perforins; (iii) blocking two targets to inhibit cancer growth; and (iv) facilitating the formation of protein complexes with antibody–drug conjugates (ADCs) [12,24] (Figure 3). 

During the immunoediting process, cancer cells develop immune escape mechanisms which allow them to evade the host’s immune response [10]. Blocking of immune checkpoints by mAbs has clearly been shown to improve cancer patients’ survival by overcoming immune escape mechanisms. bsAbs, by targeting both oncogenic signaling pathway components and immune checkpoints, have the potential to inhibit tumor cell growth and stimulate a tumor immune response simultaneously [57]. BiTEs and scFvs are the most utilized bsAbs [57]. As described above, bsAbs can be used to form a bridge between cancer cells and cytotoxic immune cells. The bridging of cancer cells and immune cells can be obtained by CD3, CD16 and CD64 expression on immune cells to engage T, natural killer and phagocytic cells, respectively [24,58]. On the other hand, cancer cells are recruited by a wide range of tumor-associated antigens (TAAs) [57]. As a result, bsAbs can lead to an enhanced recognition and lysis of cancer cells by cytotoxic immune cells or stimulate antigen-presenting cells even in the absence of MHC-I expression. In addition, immune cell activation by bsAbs also induces cytokine secretion and concomitant T-cell proliferation, sustaining an even more durable anti-cancer immune response [59].

bsAbs are also used to promote immune cell functions by targeting immune checkpoints. mAbs defined as immune checkpoint inhibitors (ICIs) are already widely utilized in oncology. Many bsAbs were developed to simultaneously target multiple immune checkpoints including programmed cell death 1 (PD-1), programmed death-ligand 1 (PD-L1) and Cytotoxic T-Lymphocyte Antigen-4 (CTLA-4) in order to enhance and re-direct a host’s immune response against cancer cells [24,60]. In addition, novel bsAbs targeting other immune checkpoints such as OX40, ICOS and CD28 were also developed [24,60]. This novel therapeutic approach is expected to enhance both therapeutic efficacy as well as toxicity as compared to mAbs.

Besides stimulating the host immune response, bsAbs are also used to exert an anti-cancer effect by simultaneously blocking two different oncogenic TAAs or two different epitopes of the same oncogenic TAA [24,60]. In the last few years, a great number of TAAs have been targeted by small molecules and consequently, a great number of bsAbs have been developed and are been testing in clinical trials [24,60]. Dual-TAA targeting is expected to enhance drug selectivity for cancer cells and ideally reduce on-target/off-tumor toxicities [24,60]. In addition, the dual receptor signaling blockade is also expected to overcome well-defined mechanisms of targeted therapy resistance [24,60]. 

Lastly, in the last decade, several types of oncological strategies have been designed to enhance drug-selective delivery. ADC is one of the most widely investigated strategies [61]. As compared to mAb-based ADC, bsAbs loaded with an ADC have the potential to improve specificity and drug internalization. As a result, bsAbs have the potential to enhance tumor cell killing and therapeutic index [19,24]. Among the different bsAbs loaded with ADC, ZW49 is one of the most investigated. ZW49 combines the property of ZW25 (described next) to deliver the N-acyl sulfonamide auristatin which is an effective anti-cancer agent [24,62]. In addition, microtubule-disrupting doustatin-3, as well as the introduction of novel effective conjugates such as toxins, radioisotopes or cytokines, represent novel potential strategies to improve the activity of this type of bsAb [24,63,64].

## 4. Clinical Evidence of Bispecific Antibodies in Solid Malignancies

Several clinical trials were implemented to test the efficacy and safety of bsAbs for the treatment of solid malignancies. In Figure 4 and Table 1, we summarized the bsAbs currently tested. In addition, several clinical trials were also implemented to investigate the potential use of bsAbs in combination with other agents including immunotherapies [12,20,24,60]. In the next section, we will provide an overview of the most promising bsAbs utilized for the treatment of solid malignancies.

### 4.1. Tumor-Associated Antigen (TAA)-Based bsAbs

#### 4.1.1. Amivantamab

Amivantamab (JNJ-61186372) is a fully human bsAb targeting both EGFR and mesenchymal–epithelial transition factor (MET) [65,66]. In Non-Small Cell Lung Cancer (NSCLC), MET amplification drives EGFR-inhibitor resistance by activating EGFR-independent phosphorylation of ErbB3 and downstream activation of the PI3K/AKT pathway. By binding both EGFR and MET, amivantamab simultaneously down-regulates EGFR and MET signaling activation through their internalization and subsequent lysosome degradation [67]. As a result, treatment with amivantamab is expected to overcome EGFR-inhibitor resistance.

In the CHRYSALIS study, 81 NSCLC patients harboring exon 20 insertions (Exon20ins) in the EGFR gene were treated with amivantamab until disease progression or unacceptable toxicity. Three complete responses were reported with an overall response rate (ORR) of 40%. The median duration of response was 11.1 months. The median progression-free survival (PFS) was 8.3 months. The most common adverse events (AEs) included rash, infusion-related reaction and paronychia (86%, 66%, and 45%, respectively). Hypokalemia was the most common grade 3-4 AE. Treatment-related dose reductions and discontinuations were reported in 13% and 4% of treated patients, respectively [68]. Based on this clinical activity, amivantamab recently received FDA approval for the treatment of NSCLC patients harboring EGFR Exon20ins following disease progression to platinum-based chemotherapy. In addition, in the combination arm of the CHRYSALIS study, forty-five NSCLC patients harboring EGFR mutations and progressed on first-line treatment with the third-generation EGFR TKI osimertinib, were treated with amivantamab in combination with the third-generation EGFR TKI lazertinib. ORR was 36%. Median PFS was 4.9 months. No sudden AEs were reported [69]. Currently, three different trials are testing the activity of amivantamab in combination with other agents in NSCLC. In the MARIPOSA trial, advanced naïve NSCLC patients carrying EGFR mutations (exon 19 or L858R) are randomized to receive amivantamab plus lazertinib, lazertinib or osimertinib. In the PAPILLON trial, advanced naïve NSCLC patients harboring EGFR exon20ins are randomized to receive platinum-based chemotherapy plus placebo or amivantamab. In the CHRYSALIS 2 trial, EGFR-mutated NSCLC patients are randomized to receive either lazertinib alone or its combination with amivantamab.

#### 4.1.2. KN026

KN026 is an Fc-based bsAb targeting two different epitopes of HER2. Specifically, it binds both the juxta-membrane extracellular domain (ECD4) and dimerization domain (ECD2) of HER2. Both these domains are targeted by the combination of HER2-specific mAbs trastuzumab and pertuzumab, currently approved for the treatment of HER2+ cancer including breast cancer [12,70]. In the NCT03619681 trial, metastatic HER2+ breast cancer patients who failed prior anti-HER2 therapy were treated with KN026. ORR and disease control rate (DCR) were 32.1% and 76.8%, respectively [71]. The treatment was well-tolerated and the most common AEs were pyrexia (23.8%), diarrhea (19.0%), aspartate aminotransferase increase (15.9%), neutrophil count decrease (11.1%) and white blood cell count decrease (11.1%) [71]. In the NCT03925974 trial, HER2+ gastric/esophageal cancer patients who have failed prior anti-HER2 were also treated with KN026. ORR and DCR, in the patient cohort overexpressing HER2 (IHC 3+ or IHC2+/ISH+), were 55.6% and 72.2%, respectively, while in the cohort of patients with low expression of HER2 (IHC 1+ or IHC 2+/ISH-, IHC 0+ or IHC1+/ISH+) were both 22.2% [72]. The overall incidence of KN026-related AEs was 87.1%; 9.7% were grade 3-4. The most common AEs were aspartate aminotransferase increase (25.8%), rash (19.4%), anemia (16.1%), alanine aminotransferase increase (12.9%) and weight decrease (12.9%). Grade 3-4 AEs included infusion-related reaction (3.2%), blood pressure increase (3.2%) and ureteral stricture with hydronephrosis (3.2%) [72]. Currently, KN026 is being tested in combination with KN046, a novel bsAb targeting both CTLA-4 and PD-1 (NCT04521179, NCT04040699). Preliminary analyses have shown a safe toxicity profile with some anti-tumor activity [73]. However, the final results are still pending.

#### 4.1.3. Zanidatamab

Zanidatamab (ZW25), a humanized IgG1 bsAb, as described for KN026, also targets the ECD4 and ECD2 domains of HER2 [74]. In the NCT02892123 trial, metastatic patients with HER2-overexpressing breast (17), gastric/esophageal (11) and other cancers (5), previously treated with anti-HER2 based therapies, received zanidatamab. Breast cancer patients included a median of six previous HER2-targeted regimens while all gastric/esophageal cancer patients received a median of four systemic therapies including trastuzumab. The DCR was 57%, 54%, and 33% in gastric/esophageal, breast and in another cancer cohort, respectively [75]. The most common AEs were diarrhea and infusion, with no AE-related discontinuations [75]. In addition, in the expansion cohort of the same study, 20 patients with HER2-overexpressing biliary tract cancer (BTC) including 11 gallbladder cancers, five intra- and four extra-hepatic cholangiocarcinomas were treated with zanidatamab. The median number of prior systemic therapies was 2.5, including five patients who had received trastuzumab. Fourteen patients (70%) experienced grade 1 or 2 AEs; the most common AEs were diarrhea (45% of patients) and infusion-related reactions (30% of patients). ORR and DCR were 47% and 65%, respectively. The median duration of response (DOR) was 6.6 months [76]. Currently, zanidatamab is being tested in (i) a phase 2b trial (NCT04466891, HERIZON-BTC-01) enrolling patients with advanced HER2-overexpressing BTC following progression to the first-line gemcitabine-based regimen, (ii) in a phase 2a trial (NCT04224272) in combination with both the cyclin-dependent kinase 4/6 inhibitor (iCDK4/6) palbociclib and the anti-hormone receptor (HR) fulvestrant in patients with advanced HER2+/HR+ breast cancer, and (iii) in a phase 1b/2 trial (NCT04276493) in combination with the anti-PD-1 tislelizumab. In the latter trial, in cohort 1, patients with advanced HER2-overexpressing breast cancer receive zanidatamab plus chemotherapy while in cohort 2, patients with advanced HER2-overexpressing gastric/esophageal cancer receive zanidatamab plus chemotherapy and tislelizumab [77].

In addition, in reason of its promising activity, the cytotoxicity of zanidatamab is also being tested as a drug-conjugated bsAb (ZW49). ZW49 is a zanidatamab-based bsAb where the bsAb is conjugated to a novel N-acyl sulfonamide auristatin payload via a protease-cleavable linker to exert a block on tubulin polymerization and cell division [62]. The activity of ZW49 is currently being investigated in a phase 1 trial (NCT03821233) enrolling HER2+ cancer patients.

#### 4.1.4. Zenocutuzumab

Zenocutuzumab (MCLA-128) is a bsAb targeting both HER2 and HER3. However, as compared to KN026 or zanidatamab or trastuzumab, zenocutuzumab targets a different epitope of HER2 [78]. As a result, zenocutuzumab is expected to overcome the HER2/HER3-mediated resistance to trastuzumab by (i) targeting a different epitope of HER2, (ii) inhibiting the interaction between HER2 and HER3; (iii) inhibiting the downstream activation of phosphoinositide 3-kinases (PI3K)-AKT signaling pathway; and (iv) inducing ADCC [79].

Zenocutuzumab was tested for the treatment of different types of solid tumors including gastric/esophageal and breast cancer [80,81,82]. In cohort 1 of the NCT03321981 trial, patients with HER2+/amplified metastatic breast cancer received zenocutuzumab following progression on HER2-specific Ab drug conjugated trastuzumab emtansine (TDM1). DCR was 77% [81]. Treatment was well tolerated with neutropenia (61%), diarrhea (61%), asthenia/fatigue (46%) and nausea (29%) as the most common AEs [81]. In cohort 2 of the same trial, patients with ER+/HER2-low metastatic breast cancer, following progression to iCDK4/6, were also treated with zenocutuzumab. DCR was 45% and a similar toxicity profile was reported for cohort 1 [81]. On the other hand, results for patients with HER2+ gastric/esophageal cancer are still pending [80].

#### 4.1.5. Vanucizumab

Neoangiogensis is essential for cancer growth and represents a well-known hallmark of cancer [83]. As a result, one of the pillars of anti-cancer therapy is based on counteracting pro-angiogenic factors such as Vascular Endothelial Growth Factor A (VEGF-A), angiopoietin 2 (Ang-2) and Delta-like Ligand 4 (DLL4) [83,84]. Vanucizumab is a bsAb targeting both VEGF-A and Ang-2 [85]. Preclinical studies demonstrated that vanucizumab strongly inhibited tumor growth and angiogenesis, with higher efficiency as compared to VEGF-A or Ang-2-based monotherapy [85]. In addition, results from different phase 1 trials yielded encouraging results [14,24]. However, vanucizumab failed to increase PFS in a phase 2 trial as compared to standard therapy. Specifically, in the NCT02141295 trial, untreated metastatic colorectal (mCRC) patients were randomized to receive mFOLFOX-6 plus vanucizumab or mFOLFOX-6 plus anti-VEGF mAb bevacizumab, as a standard of care. At a median follow-up of 17.6 months, the combination of mFOLFOX-6 plus vanucizumab did not improve PFS and was associated with a higher risk of hypertension and hemorrhagic events as compared to standard of care [86]. Further investigations are needed to clarify the efficacy of vanucizumab in mCRC as well as in other types of cancer.

### 4.2. Immune Checkpoint-Based bsAbs

#### 4.2.1. KN046

As we already discussed, ICI-based immunotherapy has revolutionized the treatment of several types of malignancies [87,88,89]. Based on these clinical findings, novel bsAbs were designed to simultaneously target two or more immune checkpoints in order to improve the host immune response against cancer cells [24,60]. On this line, KN046 was developed. KN046 is a novel IgG1 bsAb targeting both PD-L1 and CTLA-4. It prevents the binding of PD-L1 to PD-1 and of CTLA-4 to CD80/CD86 [90]. This bsAb was tested in various trials which enrolled patients with different types of malignancies [24]. In the phase 1 clinical trial NCT03529526, nasopharyngeal and NSCLC patients who progressed on previous ICI-based immunotherapy were treated with KN046. A manageable safety profile characterized by pruritus (27.6%), rash (27.6%) and asthenia (27.6%) was reported. Only 2 out of 29 treated patients experienced a grade 3-4 AEs (anemia and infusion-related reaction). ORR and DCR were 12% and 52%, respectively [91]. In addition, in the NCT03872791 trial, KN046 was tested in combination with nab-paclitaxel for the treatment of triple-negative breast cancer (TNBC). Preliminary results of this ongoing phase 2 trial demonstrated a safe safety profile characterized by aspartate and alanine aminotransferase increases (48%), pyrexia (33%), neutrophil count decrease (30%) and anemia (26%). The most common grade 3-4 AEs were neutrophil count decrease (26%), white blood cell count decrease (22%) and aspartate aminotransferase increase (15%). The median PFS was 7.33 months. The median OS was not reached. Twelve-month PFS and OS were 38.3% and 80%, respectively [92].

#### 4.2.2. Cadonilimab

Cadonilimab (AK104) is a humanized IgG1 bsAb targeting both PD-1 and CTLA-4 [12,24]. To date, this drug results to be approved by the FDA for the treatment of metastatic cervical cancer. Approval was based on the preliminary results obtained from a phase 2 trial (NCT04380805). ORR and DCR were 47.6% and 66.7%, respectively [12,24]. Many other ongoing trials are testing cadonilimab for the treatment of solid malignancies [12,24]. In patients with relapsed/refractory mesothelioma, preliminary results demonstrated that cadonilimab induced an ORR and 8-week DCR of 15.4% and 84.6%, respectively. Only 16.7% of treated patients experienced grade 3-4 AEs including fever, type 1 diabetes mellitus and infusion-related reactions; the most common grade 1-2 AEs were rash and infusion-related reactions [93]. In a phase 1 trial (NCT03852251) enrolling patients with metastatic gastric/gastroesophageal junction adenocarcinoma, cadonilimab in combination with modified chemotherapeutic XELOX regimen demonstrated ORR and DCR of 66.7% and 95.8%, respectively. The most common grade 1-2 AEs were neutropenia, thrombocytopenia, anemia, and infusion-related reaction with a percentage of 26.5%, 20.6%, 17.6% and 17.6%, respectively. The most frequent grade 3-4 AEs were both neutropenia (8.8%) and immune-related reactions (8.8%) [94]. Lastly, in a phase 2 trial (NCT04444167) enrolling patients with unresectable hepatocellular carcinoma, cadonilimab was administered in combination with TKI lenvatinib. Preliminary results demonstrated ORR and DCR of 44.4% and 77.8%, respectively. The toxicity profile included increased levels of transaminases (36.7%), thrombocytopenia (33.3%) and neutropenia (30.0%) [95].

### 4.3. Immune Cell Engagement by bsAbs

#### 4.3.1. Catumaxomab

Catumaxomab is a bsAb targeting CD3 and Epithelial Cell Adhesion Molecule (EpCAM). CD3 is expressed in T cells. EpCAM is expressed in epithelial cancer cells [96]. Catumaxomab was initially approved by the FDA for intra-peritoneal treatment of recurrent symptomatic malignant ascites in patients with EpCAM-positive cancers resistant to conventional chemotherapy. This approval was based on the results from the NCT00836654 trial, in which EpCAM-positive cancer patients were randomized to receive catumaxomab plus paracentesis (experimental group) or paracentesis alone (control group). Puncture-free survival, as well as the median time to the next paracentesis, were longer in the catumaxomab group as compared to the control group. In addition, OS analysis showed a positive trend for the experimental group [97]. Unfortunately, these promising results were not confirmed in the following trial. Indeed, patients affected by peritoneal carcinomatosis do not achieve an improvement in their survival outcomes with the addition of catumaxomab to the standard chemotherapy as compared to standard chemotherapy alone [98]. As a result, catumaxomab was subsequently withdrawn at the request of the marketing authorization holder [99].

#### 4.3.2. Ertumaxomab

Ertumaxomab is a bsAb targeting both CD3 and HER2 [100]. In addition, it can also bind some Fcγ receptor-positive immune cells forming a ternary cellular complex between cancer cells and antigen-presenting cells (APCs) [100]. In the phase 1/2 NCT01569412 trial, HER2+ cancer patients (IHC 1+/SISH positive, IHC 2+ and 3+) were treated with ertumaxomab. No dose-limiting toxicity was detected in any of the treated patients. AEs were transient and reversible [101]. No further data about the efficacy of this drug are available yet.

#### 4.3.3. MEDI-565

MEDI-565 (also known as MT111) is a bsAb that binds carcinoembryonic antigen (CEA) on cancer cells, and CD3 on T cells, to induce T-cell mediated killing of cancer cells [102]. This novel drug was developed for the treatment of patients with cancers expressing CEA. In vitro and in vivo evidence demonstrated that MEDI-565 induces T-cell-mediated killing of CEA+ cancer cells without the assistance of any co-stimulatory agents [20,103]. In the phase 1 NCT01284231 trial, thirty-nine patients with gastrointestinal adenocarcinoma were treated with MEDI-565. No objective responses were treated with MEDI-565: SD was 28%, median PFS and OS were 1.6 and 5.5 months, respectively. Five patients reported grade 3 AEs including diarrhea, cytokine release syndrome (CRS), alanine aminotransferase increased and hypertension [104]. Based on this low clinical activity, which was further confirmed in the NCT02291614 trial, despite an acceptable safety profile, the drug was discontinued [105].

#### 4.3.4. RO6958688

RO6958688 is a bsAb targeting both CEA and CD3 [106]. In contrast with MEDI-565 which binds CEA with a monovalent binding, RO6958688 harbors a bivalent binding site for CEA and a monovalent binding site for CD3 [106]. It was shown that RO6958688 efficacy strongly correlates with CEA expression, with higher efficacy observed in highly CEA-expressing cancer cells [107]. Anti-cancer activity of RO6958688 was shown in the NCT02324257 and NCT02650713 trials. In addition, RO6958688 activity appeared to be enhanced by its combination with atezolizumab, a PD-L1 specific mAb, with a manageable safety profile [108]. The updated results are still pending.

## 5. Challenges of bsAbs in Solid Malignancies

Despite the promising results emerging in numerous preclinical and clinical studies, the therapeutic potential of bsAbs for the treatment of solid malignancies has several limitations including limited biodistribution, the development of anti-drug antibodies, TME-mediated resistance mechanisms and on-target-based resistance mechanisms [24]. 

As mentioned above, Fc-free bsAbs have a short plasma half-life which is a huge drawback, especially when the antibodies must reach the solid tumor from the circulation [109]. As a result, several approaches were proposed for prolonging bsAb half-life using genetic fusion or chemical conjugation of the antibody fragment to IgG Fc domain, human serum albumin or to polyethylene glycol [28]. In addition to this strategy, Leconet et al. have developed an injectable in situ biodegradable polymer-based protein delivery system to prolong the half-life and the antitumor efficacy of BiTEs targeting both prostate-specific membrane antigen (PSMA) and the CD3 T-cell receptor in prostate cancer [110]. On the other hand, bsAbs with an Fc domain have a longer half-life in circulation as compared to Fc-free bsAbs [109]. The interaction between the Fc domain with different Fcγ-receptors on the immune cell surface can induce immune off-target effects such as ADCC, CDC, and ADCP, which can in turn increase the efficacy of the drug and/or the onset of undesired toxicities [111]. For instance, the above-described catumaxomab revealed a dose-dependent hepatotoxicity of different grades until the fulminant fatal acute liver failure. The latter was associated with the off-target binding of catumaxomab to Fcγ receptors expressed by Kupffer cells in the liver. This binding induces local cytokine release and T-cell-mediated hepatotoxicity [112]. To overcome this limitation, recently, several bsAbs with silent Fc domains have been developed by introducing point mutations that abolished the binding of Fcγ receptors to Fc domains in order to reduce or avoid ADCC and CDC [113].

Identification of TAAs, which are specifically, or at least predominantly, expressed on tumor cells rather than normal cells is another relevant hurdle that bsAbs therapy faces in solid tumors. This differential expression is needed to avoid on-target off-tumor toxicity that may be dose- and efficacy-limiting [12,114,115]. Such limitations were observed with several BiTEs which target TAAs often overexpressed in solid tumors but also expressed at lower levels in normal tissue. Identification of neoantigens, also referred to as tumor-specific antigens (TSAs), could open an opportunity to mitigate on-target off-tumor toxicity. TSAs derived from viral or mutated proteins are presented by MHC-I molecules exclusively on the surface of cancer cells and are absent on the cell surfaces of normal cells [33]. The MHC-peptide complexes can be targeted by particular bsAbs, such as immune-cell-mobilizing monoclonal TCRs against cancer molecules (ImmTAC), which include an anti-CD3 scFv linked to a high-affinity TCR that recognizes target MHC-peptide complexes [116,117]. This strategy, based on targeting intracellular TAAs, provides a broader spectrum of tumor-specific targets that are not normally accessible to antibodies, thereby reducing off-target toxicity [118,119]. Increasing the binding avidity of bsAbs, with additional antigen-binding units, represents another potential strategy to reduce off-target toxicity [113,120]. Several studies have shown that multivalency for TAAs strengthens the binding avidity of bsAbs to a tumor cell, resulting in enhanced cytotoxic potency and specificity [121,122,123]. For instance, Slaga et al. developed a CD3-bsAb that binds two HER2 molecules at the same time with low affinity, making it selective for tumors that have a high density of surface HER2 relative to healthy tissues [122]. A similarly improved selectivity was observed for RO6958688 [107]. The bivalency of the CEA confers high binding avidity to the tumor, providing better tumor targeting compared to CEA monovalent binding [107]. In addition, the use of antibodies with binding regions that are masked with protease-cleavable linkers is also expected to overcome the lack of TAA specificity [124]. The conditional activation of these prodrugs occurs in the context of the tumor, where proteases are ubiquitously expressed but not in normal tissue, where the proteolytic activity is tightly regulated [125]. On this line, Boustany et al. showed that in cynomolgus monkeys the maximum tolerated dose (MTD) for the masked EGFR-CD3-bsAb was 60-fold higher than the unmasked construct. The masked variant considerably prolonged the plasma concentrations at higher doses [126]. Lastly, intra-tumoral generation of bsAbs is also considered an approach finalized to reduce off-target toxicity. Oncolytic viruses (OVs) and genetically transduced T cells are two methods to produce bsAbs in patient tumor tissues. OVs, by selectively infecting and replicating in cancer cells, can be armed with a therapeutic transgene that encodes a BiTE, thus acting as a delivery tool for bsAbs [127,128]. These armed viruses not only cause the death of infected cancer cells through non-specific direct oncolysis but also induce localized secretion of bsAbs and T-cell-mediated bystander killing of uninfected cancer cells [127,128,129]. Endogenous secretion of T-cell redirecting bsAbs (STAb) by genetically modified T cells is also a novel strategy utilized as a delivery system for bsAb production in the TME [114,130]. Iwahori et al. generated T cells expressing a secretable T-cell engager specific for CD3 and the TAA erythropoietin-producing hepatocellular carcinoma A2 (EphA2). EphA2a is a member of the Eph family of receptor tyrosine kinases that is overexpressed in a broad range of human tumors including breast, lung, prostate, and glioblastoma [131]. This type of generated T cell displayed a potent in vivo antitumor activity in both glioma and lung cancer with a significant increase in mice survival as well as reduced systemic exposure as confirmed by the absence of human cytokines in the peripheral blood of cancer-bearing mice [131].

The lack of significant T-cell infiltration in TME is another factor limiting the therapeutic efficacy of bsAbs in solid tumors [132,133]. Immune-desert tumors are characterized by weak T-cell infiltration, thus potentially limiting the efficacy of bsAbs, especially for immune cell engager-based therapy [134]. A relevant study demonstrated that intra-tumoral injection of oncolytic reovirus into immunocompetent tumor-bearing mice induced a potent IFN response and a strong influx of T cells, thereby sensitizing the TME for subsequent CD3-bsAb therapy [135]. Combination treatment of reovirus and CD3-bsAb resulted in tumor regression, which did not occur in the absence of OV sensitization [135]. These results provide evidence that preconditioning the TME with oncolytic reovirus converts immunologically cold tumors into inflamed ones, thereby improving the efficacy of CD3-bsAb in immune-desert solid tumors [135]. Generation of bsAbs with a silenced Fc domain represents an additional potential strategy to promote T-cell infiltration in tumor sites. Wang et al. showed that bsAbs with intact Fc domains were unable to drive T cells to the tumor, thereby failing to achieve an antitumor effect in the xenograft mice model. Indeed, T cells became sequestered in the lungs by myeloid cells or depleted in circulation. In contrast, bsAbs with a silenced Fc domain enhanced T-cell infiltration as well as anti-tumor effects [113]. T-cell infiltration into TME can be limited by extracellular matrix (ECM) components. In solid malignancies, extracellular matrix (ECM) components cause the formation of a physical barrier which determines T-cell exclusion from the tumor bed, thus also limiting the efficacy of bsAbs based-therapy [136]. Cancer-associated fibroblasts (CAFs), a major player of tumor stromal cells, produce a large amount of ECM proteins such as collagens, glycol-proteins and proteoglycans [137]. Fibroblast activation protein-α (FAP) is highly overexpressed in CAFs and represents an attractive target for immunotherapy [138]. Interestingly, various studies have shown that engineered Ovs encoding CD3-bsAbs targeting FAP have the ability to kill CAFs and at the same time, increase intra-tumoral infiltration of T cells in several immunocompetent mouse models of cancer [139,140].

Tumor burden can also limit the effectiveness of bsAbs [141]. Chiu et al. showed that in humanized mouse models, efficient treatment with a PSMA- and CD3-specific bsAb required administration of a costimulatory agonistic 4-1BB in a high tumor burden condition, while no coadministration was required in a low tumor burden condition [141]. Specifically, the therapeutic combination of PSMA-CD3-bsAb with 4-1BB co-stimulation was demonstrated to induce durable antitumor responses by an efficient induction of T-cell memory and to improve the survival of mice bearing high tumor burden as compared to bsAb monotherapy [141].

The presence of a complex immune-suppressive TME also represents an important limitation for bsAbs-based therapies [142,143]. Specifically, it was observed that both PD-1 on T cells and PD-L1 on tumor cells are up-regulated during treatment with T cells engaging bsAbs, thereby limiting their activity [106,142,144]. Hettich et al. reported that a CD3-bsAb targeting the tumor stem cell marker AC133 (a stem cell-specific epitope of CD133) stimulated apoptosis of hypo-fractionated radiotherapy-induced tumor-infiltrating lymphocytes (TILs) via the PD-1 pathway. This determined the growth of melanoma tumors. Interestingly, further PD-1 blockade was related to increased TIL numbers, recovery of the efficacy of antitumor immunity, and improved survival rates [145]. In addition, other preclinical studies have reported that the anticancer efficacy of T cell-engaging bsAb treatment was significantly improved when combined with ICIs [142,143,146,147]. Collectively, all these data indicate the evidence that ICI blockade seems to play an essential role in improving the efficacy of T cell-engaging bsAb therapy. Nevertheless, some studies showed that some CD3-bsAbs targeting HER2 in breast cancer cells were insensitive to PD1/PD-L1 inhibition. As a result, the effects of PD-L1/PD-1 blocking have yet to be elucidated as it may depend on several other factors, such as the exact format of the bsAb, the target moiety used and the site of the tumor [148]. Targeting both immune checkpoints and tumor antigens can provide a strategic superiority over the combination of immune checkpoint inhibitors and bsAbs [149]. Hou et al., by generating two bsAbs (PD-1/c-Met DVD-Ig and IgG-scFv) both targeting PD-1 on T cells and MET on tumor cells, showed several advantages such as a higher specific binding capacity provided by the additional Ag-binding units and a reduction in the risk of off-target FcγR-mediated ADCC due to low affinity to Fcγ receptors [149].

Lastly, a significant challenge in the implementation of bsAb-based immunotherapy in solid tumors as well as in other types of diseases is represented by the need to mitigate the cost of production. Actually, the production of bsAbs is a highly expensive and time-consuming procedure and requires elevated expertise [150]. All these limitations might hamper the clinical implementation of efficient bsAbs. 

## 6. Conclusions

The development of bsAbs for the treatment of solid malignancies is recently experiencing a rapid expansion. The advancement in the knowledge of cancer biology and the interactions of TME allow for identifying novel potential targets, while the advancement in the technology allows for the development of more effective bsAbs, either alone or in combination with other therapeutic approaches such as immunotherapy, chemotherapy, radiotherapy and targeted therapy. The impressive preliminary results from clinical trials lead the FDA to approve amivantamab, cadonilimab, KN046, zanidatamab and zenocutuzumab for the treatment of different solid malignancies between 2020 and 2021. However, further larger phase II-III clinical trials are needed to validate these results in different types of cancers and/or settings and/or lines of therapy. In addition, there are still multiple challenges that are associated with the use of bsAb-based therapy, including pharmacokinetic issues, manufacturing difficulties, resistance mechanisms, toxicity and cost.

## Figures and Tables

**Figure 1 pharmaceutics-14-02442-f001:**
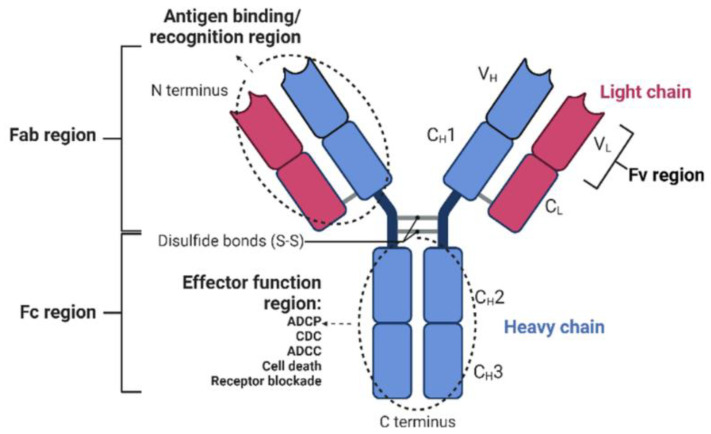
Monoclonal antibody (mAb) structure. ADCP: antibody-dependent cell phagocytosis; CDC: complement-dependent cytotoxicity; ADCC: antibody-dependent cell cytotoxicity; C: constant; V: variable; L: light chain; H: heavy chain.

**Figure 2 pharmaceutics-14-02442-f002:**
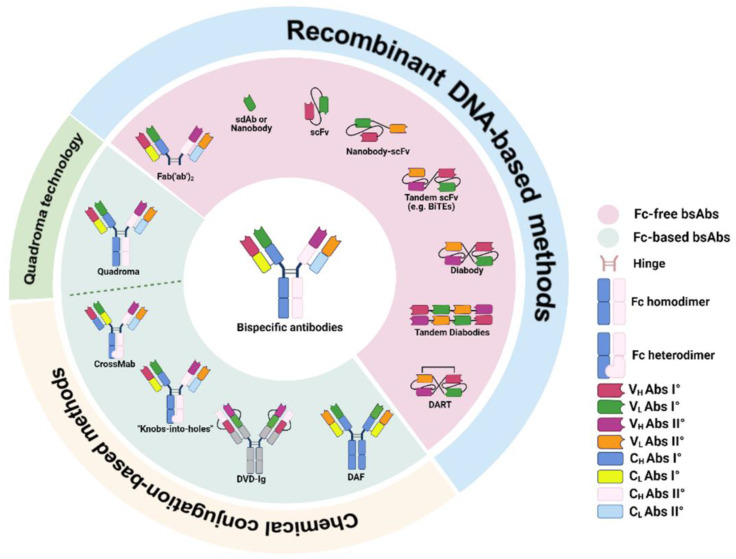
Molecular formats of bispecific antibodies. According to the Fc domain, bsAbs can be divided into Fc-free bsAb and Fc-based bsAb format molecules further classified based on their differential methods of development. Among the “zoo” of bsAbs, representation of bsAbs described in the text is shown.

**Figure 3 pharmaceutics-14-02442-f003:**
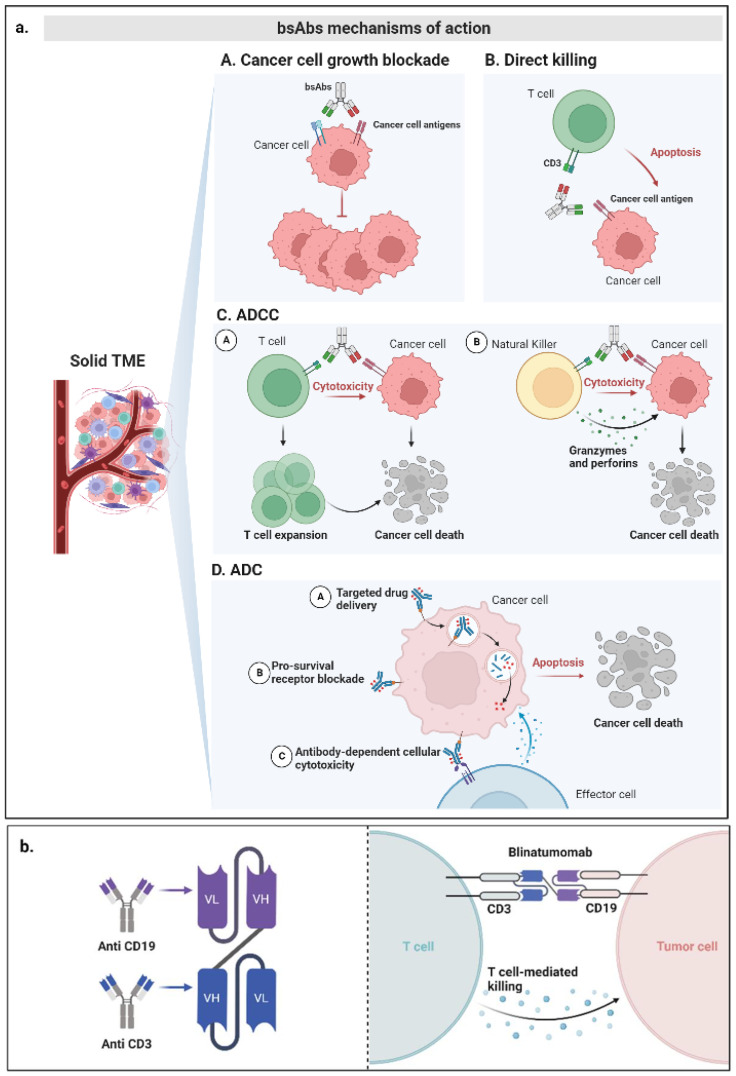
Targeting effector cells with bsAbs for cancer therapy. (**a**) Illustration of bsAb mechanisms of action. Binding of bsAbs to their targets initiates downstream signaling pathway activation that results in (**A**) cancer cell growth blockade, (**B**) cancer cell apoptosis induction by cytotoxic T cells, (**C**) cancer cell apoptosis induction by ADCC (Antibody-Dependent Cell Cytotoxicity) followed by T-cell expansion (panel on the left) as well as by granzyme and perforin release from Natural Killer cells (panel on the right), (**D**) cancer cell apoptosis induction by ADC (Antibody–Drug Conjugated) which mediates target drug delivery (up illustration), pro-survival receptor blockade (middle illustration) and ADCC (bottom illustration). (**b**) Structure and mechanism of action of Blinatumomab. TME: Tumor microenvironment.

**Figure 4 pharmaceutics-14-02442-f004:**
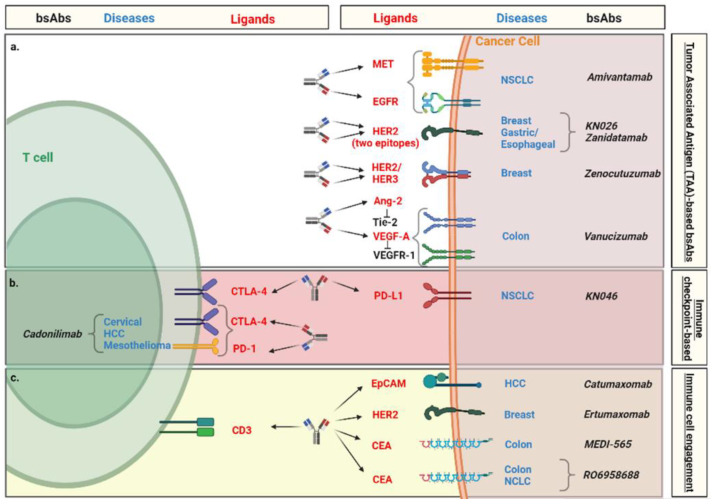
bsAbs currently tested for the treatment of solid malignancies. bsAbs are grouped into three major categories based on the biological targets and mechanism of action. (**a**) Tumor-associated antigen (TAA)-based bsAbs. (**b**) Immune checkpoint-based bsAbs. (**c**) Immune cells engage antigen-based bsAbs. Differential font colors are used to illustrate ligand (red), disease (blue) and name of bsAb (black).

**Table 1 pharmaceutics-14-02442-t001:** Current clinical trials of bsAbs based on mechanism of action in solid malignancies.

Agents	Format	Targets	Cancer	Trial Phase	NCT Number	Status	Company
**1. Tumor Associated Antigen (TAA)-Based bsAbs**							
Amivantamab	Quadroma	c-Met/EGFR	NSCL	1	NCT02609776	R	Janssen Research & Development, LLC (Raritan, NJ, USA)
			Solid tumors	1	NCT04606381	R	
			NSCL	1	NCT04077463	R	
			UGI	2	NCT04945733	R	
			NSCL	3	NCT04538664	R	
			NSCL	3	NCT04487080	ANR	
KN026	CrossMab	HER2	Breast/UGI	1	NCT03619681	ANR	Jiangsu Alphamab Biopharmaceuticals Co., Ltd. (Suzhou, China)
			Breast/UGI	1	NCT03847168	ANR	
			Breast	2	NCT04881929	R	
			GI	2	NCT03925974	U	
			Solid tumors	2	NCT04521179	ANR	
			Breast	2	NCT04778982	R	
			Breast	2	NCT04165993	ANR	
Vanucizumab	CrossMab	Ang-2/VEGF-A	Solid tumors	1	NCT02665416	C	Hoffmann-La Roche (Basel, Switzerland)
			Solid tumors	1	NCT01688206	C	
			Solid tumors	1	NCT02715531	C	
			CR	2	NCT02141295	T	
Zanidatamab	CrossMab	HER2	Solid tumors	1	NCT02892123	ANR	Zymeworks Inc./BeiGene, Ltd. (Vancouver, BC, Canada)
			Breast/UGI	1/2	NCT04276493	ANR	
			Biliary	2	NCT04466891	ANR	
			Breast	2	NCT04224272	R	
			GI	2	NCT03929666	R	
Zenocutuzumab	CrossMab	HER2/HER3	Solid tumors	1/2	NCT02912949	R	Merus N.V. (Utrecht, The Netherlands)
			Breast	2	NCT03321981	ANR	
**2. Antibody Drug Conjugates (ADC)-Based bsAbs**							
ZW49	CrossMab	HER2	Solid tumors	1	NCT03821233	R	Zymeworks Inc. (Vancouver, BC, Canada)
**3. Immune Checkpoint-Based bsAbs**							
Cadonilimab	IgG-scFv	CTLA-4/PD-1	Solid tumors	1	NCT03261011	U	Akeso|Akeso Pharmaceutical, Inc. (Hong Kong, China)
			Solid tumors	1	NCT04572152	R	
			NSCL	1/2	NCT04647344	NYR	
			Solid tumors	1/2	NCT03852251	U	
			Solid tumors	1/2	NCT04172454	U	
			HC	1/2	NCT04444167	U	
			NSCL	1/2	NCT04646330	ANR	
			NSCL	2	NCT04544644	NYR	
			HC	2	NCT04728321	R	
			GI	2	NCT04556253	NYR	
			Cervical	2	NCT04380805	ANR	
			Solid tumors	2	NCT04547101	R	
			NF	2	NCT04220307	U	
			Cervical	2	NCT04868708	ANR	
KN046	Quadroma	CTLA-4/PD-L1	Solid tumors	1	NCT03529526	U	Jiangsu Alphamab Biopharmaceuticals Co., Ltd. (Suzhou, China)
			Solid tumors	1	NCT03733951	R	
			Solid tumors	1	NCT04040699	R	
			Breast	1/2	NCT03872791	ANR	
			GI	1/2	NCT04612712	R	
			HC	1/2	NCT04601610	ANR	
			Thymic	2	NCT04925947	R	
			Thymic	2	NCT04469725	R	
			UGI	2	NCT03925870	U	
			UGI	2	NCT03927495	U	
			NSCL	2	NCT03838848	U	
			NSCL	2	NCT04054531	U	
			Breast	2	NCT04165993	ANR	
			HC	2	NCT04542837	R	
			Solid tumors	2	NCT04521179	ANR	
			NSCL	3	NCT04474119	ANR	
**4. Immune Cell Engagement by bsAbs**							
Catumaxomab	Qaudroma	CD3/EpCAM	Solid tumors	1	NCT01320020	T	Neovii Biotech/ Fresenius Biotech NorthAmerica (Boston, MA, USA)
			Bladder	1	NCT04819399	R	
			Bladder	1/2	NCT04799847	R	
			Ovarian	2	NCT00189345	C	
			Ovarian	2	NCT00377429	C	
			Ovarian	2	NCT01815528	C	
			Ovarian	2	NCT01246440	C	
			Ovarian	2	NCT00563836	C	
			Gastric	2	NCT00464893	C	
			Gastric	2	NCT00352833	C	
			Gastric	2	NCT01784900	T	
			GI	2	NCT01504256	C	
			MA	2	NCT01065246	C	
			MA	2	NCT00326885	C	
			MA	2/3	NCT00836654	C	
			MA	3	NCT00822809	C	
			Gastric	3	NCT04222114	R	
Ertumaxomab	Quadroma	CD3/HER2	Solid tumor	1/2	NCT01569412	T	Neovii Biotech/ Fresenius Biotech North America (Boston, MA, USA)
			Breast	2	NCT00452140	T	
			Breast	2	NCT00351858	T	
			Breast	2	NCT00522457	T	
MEDI-565	BITE	CD3/CEA	GI	1	NCT01284231	C	MedImmune LLC (Gaithersburg, MD, USA)
			GI	1	NCT02291614	T	
RO6958688	CrossMab	CD3/CEA	Solid tumors	1	NCT02324257	C	Hoffmann-La Roche (Basel, Switzerland)
			Solid tumors	1	NCT02650713	C	
			CR	1	NCT03866239	ANR	
			CR	1/2	NCT04826003	R	
			NSCL	1/2	NCT03337698	R	

Abbreviations: Ang-2: angiopoietin 2; bsAb: bispecific antibody; ANR: Active, not recruiting; C: completed; CD3: cluster of differentiation 3; CEA: Carcino-Embryonic Antigen; cMet: C-mesenchymal-epithelial transition factor; CR: Colorectal; CTLA-4: Cytotoxic T-Lymphocyte Antigen-4; EGFR: Epidermal Growth Factor Receptor; EpCAM: Epithelial Cell Adhesion Molecule; GI: Gastrointestinal; HC: hepatocellular; HER2: Human Epidermal Growth Factor Receptor 2; HER3: Human Epidermal Growth Factor Receptor 3; MA: Malignant Ascite; NSCL: Non-Small Cell Lung; NYR: Not yet recruiting; PD-L1: programmed death-ligand 1; PD-1: programmed cell death 1; R: Recruting; TAA: Tumor-Associated Antigen; T: Termined; UGI: Upper Gastrointestinal; U: Unknown; VEGF-A: Vascular Endothelial Growth Factor A.

## Data Availability

Not applicable.

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
