# Peer review of "Bispecific Antibodies: A Novel Approach for the Treatment of Solid Tumors"

_pharmaceutics, 2022, doi:10.3390/pharmaceutics14112442_

Round 1

Reviewer 1 Report (New Reviewer)

The authors have reviewed on the topic of bispecific antibodies for the treatment of solid tumors. The review is interesting but still requires improvement in the following aspects:

1. Need to add more figures describing the mechanism of bsAb on tumor 

2. Author states that bsAb is already approved by FDA, further saying that Phase II and III clinical data are awaited, which is confusing.

3. Cost analysis, therapeutic cost of bsAb and toxicity reports may improve manuscript quality.

Author Response

1. Need to add more figures describing the mechanism of bsAb on tumor.

Response: We added new figures describing the mechanism of bsAb on tumor

2. Author states that bsAb is already approved by FDA, further saying that Phase II and III clinical data are awaited, which is confusing.

Response: To avoid the confusion, we specified in the text that further clinical data are needed to validate bsAb efficacy in different types of cancer and/or setting and/or line of therapy.

3. Cost analysis, therapeutic cost of bsAb and toxicity reports may improve manuscript quality.

Response: We have highlighted in the paragraph “Challenges of bsAbs in Solid Malignancies” that manufacturing of bsAbs is a very expensive procedures as well as time consuming and requires an elevated expertise limiting the clinical implementation of bsAbs. Toxicity profiles have been described for the clinically tested bsAbs reported.   

Reviewer 2 Report (New Reviewer)

The manuscript "Bispecific Antibodies: Novel Approach for the Treatment of Solid Tumors" submitted by Francesco Sabbatino et al. is devoted to in-depth overview about monoclonal antibodies (mAbs) used in cancer therapy.

This manuscript, provide an overview about the state of the art of bispecific antibodies (bsAbs) for the treatment of solid malignancies outlining their classification, design, main technologies utilized for production, mechanisms of action, updated clinical evidence and potential limitations. This work is valuable and inspires further research on bispecific antibodies or drugs conjugated with mAbs for dual action cancer therapy. The work is presented clearly and answers the questions posed.

I recommend that this manuscript be accepted for publication after minor revision.

I have a few comments:

1.     Authors are kindly requested to write correctly the sentence "Dual variable domain immunoglobulin (DVD-Ig ™)" - "Dual Variable Domain Immunoglobulin (DVD-Ig ™)".

2.     Authors are kindly requested to write correctly the sentence “Knobs-into-holes (KiH)” - Knobs-into-Holes (KiH)

3.    Line 217 - “I Involvement of the Fc region” - please write the sentence correctly

4.    Line 382 – Please change word “everal” to “several”

5.     Please remove the double spaces throughout the manuscript

6.     The authors of the paper in the review mentioned the combination of antibodies with various chemotherapeutic agents.

In recent years, there has been an intense development of therapeutic nucleic acids (TNA) also in cancer treatment. Different therapeutic strategies are used, including antisense oligonucleotides (ASO) or small interfering RNA (siRNA) targeting different genes: EGFR, AXL, STAT-3. In recent work, several TNA-conjugated antibodies have been studied in detail (doi: https://doi.org/10.1038/d41573-021-00213-5 , https://doi.org/10.3390/molecules22091393 , https://doi.org/10.3390/jcm10040838 , https://doi.org/10.1021/acs.bioconjchem.9b00306; 10.1158/1078-0432.CCR-18-1277). I suggest the authors to supplement the manuscript with interesting future field on TNA-conjugated mAbs that focus on inhibiting genes: EGFR, AXL, STAT-3 in the treatment of cancer.

Author Response

  1. Authors are kindly requested to write correctly the sentence "Dual variable domain immunoglobulin (DVD-Ig ™)" - "Dual Variable Domain Immunoglobulin (DVD-Ig ™)".
  2. Authors are kindly requested to write correctly the sentence “Knobs-into-holes (KiH)” - Knobs-into-Holes (KiH)
  3. Line 217 - “I Involvement of the Fc region” - please write the sentence correctly
  4. Line 382 – Please change word “everal” to “several”
  5. Please remove the double spaces throughout the manuscript

Response: The manuscript has been carefully checked and revised.

6. The authors of the paper in the review mentioned the combination of antibodies with various chemotherapeutic agents. In recent years, there has been an intense development of therapeutic nucleic acids (TNA) also in cancer treatment. Different therapeutic strategies are used, including antisense oligonucleotides (ASO) or small interfering RNA (siRNA) targeting different genes: EGFR, AXL, STAT-3. In recent work, several TNA-conjugated antibodies have been studied in detail (doi: https://doi.org/10.1038/d41573-021-00213, https://doi.org/10.3390/molecules22091393 , https://doi.org/10.3390/jcm10040838 , https://doi.org/10.1021/acs.bioconjchem.9b00306; 10.1158/1078-0432.CCR-18-1277). I suggest the authors to supplement the manuscript with interesting future field on TNA-conjugated mAbs that focus on inhibiting genes: EGFR, AXL, STAT-3 in the treatment of cancer.

Response: We agree with reviewer 2 that in the past few years there has been an intense development of therapeutic nucleic acids (TNA) also in cancer treatment. Preclinical results of this novel cancer therapeutic approach are very interesting. However, to our knowledge no clinical data are available for conjugated TNA. Indeed, only TNA-conjugated monoclonal antibodies have been developed, so far. In addition, the development of conjugated bsAbs is at an early phase as compared to other types of bsAbs and more time is needed to potentially developed TNA-conjugated bsAbs. As a result, implementation of TNA-based therapy for cancer therapy is not a focus of our work.            

Round 2

Reviewer 1 Report (New Reviewer)

Accept

This manuscript is a resubmission of an earlier submission. The following is a list of the peer review reports and author responses from that submission.

Round 1

Reviewer 1 Report

The manuscript entitled “Bispecific antibodies: a novel approach for the treatment of solid tumors” by Liguori L, Polcaro G et al. summarized recent progress on bispecific antibody as a novel modality to treat solid tumors. The manuscript discussed the classification, design and production, mechanism, current clinical trial progress as well as challenges of bispecific antibodies. The manuscript is well structured with clear logic and comprehensive discussion. Following revisions are needed before considered for publishing:

1. Under “Introduction” section, the discovery and development history of bispecific antibody should be expanded more. The authors could consider including how bispecific antibody field evolves from first concept come-up to extensive exploration to first drug approval by the time.

2. In Figure2, the authors nicely outlined different formats of bispecific antibodies with graphic diagram. However, the difference of label between “Fc homodimer” and “Fc heterodimer” is too minor to be noticed. It should be revised to better distinguish them in the graph.

3. In Figure4, the title column above graph seems not totally agree with the graph content. Please revise it or consider separating diagrams of these three categories to avoid the confusion.

4. Table 1 well summarized ongoing clinical trials using bsAbs in solid malignancies. It’ll be better if the authors can add “molecular format” and clinical trial filed “company” info into the table as well as main text to provide an all-side review of bsAbs drugs in clinical trial.

Author Response

We want to thank the  Reviewers for his/her constructive comments and suggestions. We have revised our manuscript. Here the list of responses to the  Reviewer comments. 

The manuscript entitled “Bispecific antibodies: a novel approach for the treatment of solid tumors” by Liguori L, Polcaro G et al. summarized recent progress on bispecific antibody as a novel modality to treat solid tumors. The manuscript discussed the classification, design and production, mechanism, current clinical trial progress as well as challenges of bispecific antibodies. The manuscript is well structured with clear logic and comprehensive discussion. Following revisions are needed before considered for publishing:

Comment: Under “Introduction” section, the discovery and development history of bispecific antibody should be expanded more. The authors could consider including how bispecific antibody field evolves from first concept come-up to extensive exploration to first drug approval by the time.

Response: We thank the Reviewer for his/her comments. According to Reviewer’s suggestion, we have expanded the introduction section adding a brief summary about the discovery and the development history of bispecific antibody.

Comment: In Figure2, the authors nicely outlined different formats of bispecific antibodies with graphic diagram. However, the difference of label between “Fc homodimer” and “Fc heterodimer” is too minor to be noticed. It should be revised to better distinguish them in the graph.

Response: We have revised the Figure2 according to Reviewer’s suggestion.

Comment: In Figure4, the title column above graph seems not totally agree with the graph content. Please revise it or consider separating diagrams of these three categories to avoid the confusion.

Response: We have revised the Figure4.

Comment: Table 1 well summarized ongoing clinical trials using bsAbs in solid malignancies. It’ll be better if the authors can add “molecular format” and clinical trial filed “company” info into the table as well as main text to provide an all-side review of bsAbs drugs in clinical trial.

Response:  We thank the Reviewer to give us the opportunity to improve the quality of our review. We have revised the Table1 and the text adding molecular format and corresponding company names where antibodies are made.

Reviewer 2 Report

The authors review literature describing therapeutic bispecific antibodies. Overall, the review is very general touching upon several aspects related to bispecifics and lacks focus on a particular topic of interest. My major concern is that the authors need to indicate how this review is different from several other existing reviews on bispecific antibodies. This should be stated explicitly in the introduction. In addition, I have the below comments for the authors.

1.     The below mentioned has been stated in the abstract. While that is one reason, there are other reasons for developing bispecifics, some of which have been discussed in the body of this review as well. The authors should reword the sentence to make that clear. 

“To overcome this off-target resistance, bispecific antibodies (bsAbs) have been developed to simultaneously target differential oncogenic pathway components including those involved in cancer-immune cell interaction.”

2.     The below mentioned has been stated in introduction (line 48). The statement is misleading as Fab doesn’t always block pathway transduction. Fabs may have agonistic or antagonistic or neutral properties which is also mentioned in other parts of the review. 

“Fab region is responsible for binding to unique epitopes [4], blocking the corresponding downstream signalling pathway, while Fc region interacts with Fc receptor expressed on immune cells and promotes various effector functions [5,6].”

3.     Section 2 on classification of bispecifics is unorganized and over simplified. It is also strange that the authors seem to suggest subclassification based on valency within Fc based bispecifc antibodies. This is misleading as bispecific fragments may also have different valences based on the format. This section lacks sufficient details on formats and classification, seems disconnected.

4.     Figure 2 is not helpful and shows only a subset of bispecific antibody formats. Several other formats exist that are not shown or discussed. The authors need to acknowledge that and rephrase the figure legend to clarify the same.

5.     The categorization of bispecifics seems odd. TAA is involved in both Fig 4a and 4b, while immune cells are involved in both 4b and 4c.

6.     Table 1 looks incomplete. Including additional details such as molecule format, MOA, company name etc will make it informative. Also, it is unclear why authors focused on selected handful of examples in this table, and corresponding text in section 5. If there was a rationale for picking these examples, the authors should state the same. Several other bispecifics targeting other targets/indications that are in clinical testing are missing from this table and text e.g., PSMA/prostate, gp100/melanoma etc.

7.     While section 5 is somewhat informative, the other sections seem to be superficial without adequate details.

8.     Several statements in the review are unclear and need to be rephrased for clarity. Below are a few examples.

“To date, this drug results to be approved by FDA for the treatment of metastatic cervical cancer. Approval was reached based on the preliminary results obtained from a phase 2 trial (NCT04380805).” (line 557)

“Lastly, production of bsAbs in the tumor regions is also considered an approach finalized to reduce off-target toxicity.” (line 677)

Author Response

We want to thank  Reviewer for his/her constructive comments and suggestions. We have revised our manuscript. Here the list of responses to the  Reviewer comments. 

Comment: The authors review literature describing therapeutic bispecific antibodies. Overall, the review is very general touching upon several aspects related to bispecifics and lacks focus on a particular topic of interest. My major concern is that the authors need to indicate how this review is different from several other existing reviews on bispecific antibodies. This should be stated explicitly in the introduction.

Response: We thank the Reviewer for sharing his point of view about our manuscript. The focus of our review is represented by an up-to-date of main clinical trials in which bispecific antibodies have been tested or are being tested for the treatment of solid malignancies. We have modified the introduction in order to explicit the focus of our review. In addition, we aim to provide a global overview of types of bsAb classification, design, production methods, mechanisms of action, and clinical staging of efficacy evaluation.

Comment: In addition, I have the below comments for the authors.  The below mentioned has been stated in the abstract. While that is one reason, there are other reasons for developing bispecifics, some of which have been discussed in the body of this review as well. The authors should reword the sentence to make that clear. 

“To overcome this off-target resistance, bispecific antibodies (bsAbs) have been developed to simultaneously target differential oncogenic pathway components including those involved in cancer-immune cell interaction.”

Response: The sentence has been revised.

Comment: The below mentioned has been stated in introduction (line 48). The statement is misleading as Fab doesn’t always block pathway transduction. Fabs may have agonistic or antagonistic or neutral properties which is also mentioned in other parts of the review. 

“Fab region is responsible for binding to unique epitopes [4], blocking the corresponding downstream signalling pathway, while Fc region interacts with Fc receptor expressed on immune cells and promotes various effector functions [5,6].”

 Response: The sentence has been revised.

Comment: Section 2 on classification of bispecifics is unorganized and over simplified. It is also strange that the authors seem to suggest subclassification based on valency within Fc based bispecifc antibodies. This is misleading as bispecific fragments may also have different valences based on the format. This section lacks sufficient details on formats and classification, seems disconnected.

Response: We thank the Reviewer to give us the opportunity to improve the quality of our review. According with Reviewer’s suggestions, we reorganized the classification of bispecific antibodies.

Comment: Figure 2 is not helpful and shows only a subset of bispecific antibody formats. Several other formats exist that are not shown or discussed. The authors need to acknowledge that and rephrase the figure legend to clarify the same.

Response: We thank the Reviewer for this important suggestion. We have revised the manuscript according to the reviewer comment underlining that a “zoo” exits for bsAbs. In addition we have modified the description of the figure legend to acknowledge that.

Comment: The categorization of bispecifics seems odd. TAA is involved in both Fig 4a and 4b, while immune cells are involved in both 4b and 4c.

Response: Figure 4 has been reorganized.

Comment: Table 1 looks incomplete. Including additional details such as molecule format, MOA, company name etc will make it informative. Also, it is unclear why authors focused on selected handful of examples in this table, and corresponding text in section 5. If there was a rationale for picking these examples, the authors should state the same. Several other bispecifics targeting other targets/indications that are in clinical testing are missing from this table and text e.g., PSMA/prostate, gp100/melanoma etc.

Response: We thank the Reviewer to give us the opportunity to improve the quality of our review. We have improved the Table1 adding molecular format, corresponding company names of clinical trials and mechanism of action (MOA). The table represents a graphical overview of the bsAbs described in the text. As we stated for comment 1, represented and described bsAbs have been selected among the “zoo” of available bsAbs based on bsAb classification, design, production methods, mechanisms of action, and clinical staging of efficacy evaluation. We have better specified it in the manuscript.

Comment: While section 5 is somewhat informative, the other sections seem to be superficial without adequate details.

Response: We have modified and reorganized bsAb classification to improve our review. As stated for comment 1, our aim was to provide a global overview of types of bsAb classification, design, production methods, mechanisms of action, and clinical staging of efficacy. Although, we agree with the Reviewer’s comment about the lack of a deep description of details in other sections out of the fifth, in order to facilitate the read of our review, we have summarized the information available in the literature to provide informative and adequate sections.

Comment: Several statements in the review are unclear and need to be rephrased for clarity. Below are a few examples.

“To date, this drug results to be approved by FDA for the treatment of metastatic cervical cancer. Approval was reached based on the preliminary results obtained from a phase 2 trial (NCT04380805).” (line 557)

“Lastly, production of bsAbs in the tumor regions is also considered an approach finalized to reduce off-target toxicity.” (line 677)

Response: The manuscript has been carefully checked and revised according to Reviewer’s suggestion.

Round 2

Reviewer 2 Report

While the authors made an attempt to address some of the comments, the major concerns raised are still not adequately addressed. As indicated earlier, the review is still very general touching upon various aspects related to bispecifics and lacks emphasis on any particular topic of interest. The justification provided by the authors does not sufficiently address how this review is different from several other published reviews on bispecifics and what it adds to the ample literature available on bispecifics. The changes made do not adequately address these concerns.

Author Response

We regret to be informed that the manuscript is still not acceptable in the present form based on the reviewer comments. As we stated in the previous round of reply to the reviewer comments we have tried to provide an overview and update on most clinically relevant bispecific antibodies currently tested in clinical trials. We are aware that many and many other bispecific antibodies are currently designed, developed and implemented in clinical trials. However we have tried to summerize them by an objective view  as a clinical relevant point. Based on the other reviewer the manuscript is defined well structured and sound. Therefore we do not agree with the general comments made by the reviewer 2. To be specific, the first part of the current review is a description of transition from monoclonal antibodies to bispecific antibodies as well as classification and production of Bispecific Antibodies. The second part is related to the mechanisms of action of bispecific antobodies. The third part is related to clinical activity of the antibodies described in the first part. It appears not clear what the reviewer refers to that no emphasis is present. Emphasis cannot be present into description of classification or production as well as into description of mechanims of action. in addiiton when describing clinical activity we just objectively reported the clinical evidences available in the literature as they have been described. Finally, the last part of the manuscript describes the Challenges of bsAbs in Solid Malignancies. In this part we  emphasize the most clinically relevant limitations. For all these reasons we do not agree with the reviewer arguing that his comments are too general and do not appear to be appropriate.